Resource

# Gene editing enables T-cell engineering to redirect antigen specificity for potent tumor rejection

Julian J Albers[1], Tim Ammon[2], Dario Gosmann[1], Stefan Audehm[1], Silvia Thoene[3,4], Christof Winter[3,4] , Ramona Secci[3], Anja Wolf[5], Anja Stelzl[1], Katja Steiger[6,4], Jürgen Ruland[3,4,7], Florian Bassermann[1,4], Christian Kupatt[5,8], Martina Anton[9], Angela M Krackhardt[1,4]

Adoptive transfer of TCR transgenic T cells holds great promise for treating various cancers. So far, mainly semi-randomly integrating vectors have been used to genetically modify T cells. These carry the risk of insertional mutagenesis, and the sole addition of an exogenous TCR potentially results in the mispairing of TCR chains with endogenous ones. Established approaches using nonviral vectors, such as transposons, already reduce the risk of insertional mutagenesis but have not accomplished site-specific integration. Here, we used CRISPR-Cas9 RNPs and adeno-associated virus 6 for gene targeting to deliver an engineered TCR gene specifically to the TCR alpha constant locus, thus placing it under endogenous transcriptional control. Our data demonstrate that this approach replaces the endogenous TCR, functionally redirects the edited T cells' specificity in vitro, and facilitates potent tumor rejection in an in vivo xenograft model.

## Introduction

In recent years, the adoptive transfer of genetically reprogrammed T cells has gained more and more momentum (Lim and June 2017; June & Sadelain, 2018). Although the use of T cells expressing a chimeric antigen receptor (CAR T cells) has already advanced to the clinic, the adoptive transfer of TCR transgenic cells (TCR T cells) is less far developed (Morgan et al, 2006; Rapoport et al, 2015; Tran et al, 2016). Nonetheless, TCR T cells hold promise for also targeting intracellular antigens and thereby greatly enlarge the scope of potentially targetable antigens (Harris & Kranz, 2016), including neoantigens which are very attractive targets for personalized

tumor therapy (Bassani-Sternberg et al, 2016; Stronen et al, 2016; Zacharakis et al, 2018).

A main challenge in engineering TCR T cells is the genetic modification of primary T cells. Previously, lenti-, γ-retroviruses or nonviral vectors, such as transposons, were the vector of choice for stable integration of the tumor-reactive TCR (Morgan et al, 2006; Peng et al, 2009; Robbins et al, 2011; Rapoport et al, 2015; Rosenberg & Restifo, 2015; Deniger et al, 2016; Clauss et al, 2018). Although integrating viral vectors are in clinical use, they carry the risk of insertional mutagenesis (Hacein-Bey-Abina et al, 2003). With the advent of individualized cellular immunotherapies that target patient-specific antigens, it is also essential to produce personalized vectors, which will further increase the effort needed for clinical translation (Krackhardt et al, 2018). Nonviral vectors such as transposons are easier to implement clinically and show a more favorable integration pattern, but also only integrate in a nonspecific fashion (Tipanee et al, 2017).

Another problem with these approaches is that they only add an additional TCR gene to the already existing, endogenous one instead of replacing it (Bendle et al, 2010; van Loenen et al, 2010). This poses a problem as the introduced TCR competes with the endogenous TCR for CD3-binding sites and thus for surface expression (Ahmadi et al, 2011). The introduced TCR chains could also potentially bind endogenous chains, leading to new TCRs with potentially hazardous specificities (Bendle et al, 2010; van Loenen et al, 2010). Several engineering strategies were developed to avoid these problems by increasing the specific binding of the introduced TCR chains and to improve surface expression of the introduced TCR. These include, for instances, codon optimization (Scholten et al, 2006), the usage of a self-cleaving 2A peptide for equimolar expression (Leisegang et al, 2008), the exchange of the constant regions of the α- and β chains for murine sequences (Cohen et al, 2006) and the introduction of additional cysteine bonds (Kuball

[1]Klinik und Poliklinik für Innere Medizin III, Klinikum Rechts der Isar, Technische Universität München, Munich, Germany   [2]Experimental Hematology Group, Klinik und Poliklinik für Innere Medizin III, Klinikum Rechts der Isar, Technische Universität München, Munich, Germany   [3]Institut für Klinische Chemie und Pathobiochemie, Klinikum Rechts der Isar, Technische Universität München, Munich, Germany   [4]German Cancer Consortium (DKTK), Partner-site Munich and German Cancer Research Center (DKFZ), Heidelberg, Germany   [5]Klinik und Poliklinik für Innere Medizin I, Klinikum Rechts der Isar, Technische Universität München, Munich, Germany   [6]Institut für Allgemeine Pathologie und Pathologische Anatomie, Technische Universität München, Munich, Germany   [7]German Center for Infection Research (DZIF), Partner Site Munich, Munich, Germany   [8]German Center for Cardiovascular Research (DZHK), Partner-Site Munich Heart Alliance, Munich, Germany   [9]Institut für Molekulare Immunologie und Experimentelle Onkologie und Therapieforschung, Klinikum Rechts der Isar, Technische Universität München, Munich, Germany

Correspondence: angela.krackhardt@tum.de

et al, 2007). Additional strategies include swapping domains between the α- and β chain, adding a leucine fusion protein, or using single-chain TCRs (Govers et al, 2010; Knies et al, 2016; Foley et al, 2017). Other approaches focus on the endogenous TCR and aim to reduce its expression by disrupting the TCR gene with nucleases (Provasi et al, 2012; Legut et al, 2018) or silencing it with miRNAs (Clauss et al, 2018).

CRISPR-Cas9 has been proven to be an efficient method for gene disruption in primary human T cells (Osborn et al, 2016; Knipping et al, 2017; Seki & Rutz, 2018) and is already being used in clinical trials (NCT03399448 [ClinicalTrials.gov 2018]). In fact, targeted integration of a CAR into the TCR alpha constant (TRAC) locus was shown to potently redirect T cells against new antigens (Hale et al, 2017; MacLeod et al, 2017). Furthermore, placing the transcription of the transgene under the control of the endogenous promoter enhanced in vivo tumor control in mice (Eyquem et al, 2017). Just recently, it was demonstrated that co-electroporation of RNPs and double-strand DNA (dsDNA) enabled targeted integration of a shortened TCR construct into the TRAC locus. The introduced TCR construct lacks the constant region of the TCR α chain and uses the endogenous promoter and the endogenous α constant region when correctly integrated (Roth et al, 2018). Although this design elegantly illustrates the possibilities of targeted integration, it relies on the endogenous TRAC sequence and thus hinders TCR engineering strategies modifying this region of the introduced TCRs.

Here, we used CRISPR-Cas9 RNPs and adeno-associated viruses (AAV6) to site specifically integrate a 2.3-kb-long TCR construct into the TRAC locus, thereby replacing the endogenous TCR. By using a codon-optimized, complete TCR construct with murine constant regions and an additional disulfide bond, we were able to combine the advantages of engineered TCR constructs with those of the targeted integration of the transgene.

Our data show that targeting a TCR to the TRAC locus and placing it under the transcriptional control of the endogenous regulatory network redirects the specificity of the modified T cells and enables them to specifically eliminate tumor cells in vitro and in a murine in vivo tumor xenograft model.

## Results

### Targeted integration of a TCR into the *TRAC* locus

To induce a double-strand break in the gene encoding the TCR α chain, we designed a gRNA targeting the first exon of the TRAC locus. This region is specifically attractive as it is shared between all rearranged T cells, and a disruption in the first exon is located upstream of the functional region needed for surface expression (Eyquem et al, 2017). CRISPR-Cas9 RNPs were used to induce the double-strand break as they were shown to be a highly efficient delivery method of CRISPR-Cas9 for primary human T cells (Schumann et al, 2015; Seki & Rutz, 2018). Flow cytometric analysis of the cells showed an average knockout efficiency of 51% (Fig 1A). The knockout was confirmed by Droplet Digital PCR (ddPCR) (Mock et al, 2016), which quantified the gene-editing frequency of TRAC alleles as 40% using 10 ng genomic DNA input (Fig 1B and C). Using 100 ng

genomic DNA input, the gene-editing frequency was 47%, which is in line with the flow cytometric analysis (Fig S1).

Next, we designed a targeting construct to knock-in a TCR into the TRAC locus via HDR. For this, we used the previously described TCR2.5D6 (Klar et al, 2014). It was shown to recognize a myeloperoxidase-derived peptide, representing a tumor-associated antigen in patients with myeloid neoplasias, when presented on HLA-B7. The TCR construct was designed as a promoter trap to capture the endogenous promoter of the TRAC locus when it correctly integrates, thereby omitting the need for exogenous regulatory elements that risk insertional mutagenesis (Hacein-Bey-Abina et al, 2003). Furthermore, the TCR construct has a codon-optimized sequence, murine constant regions, and an additional disulfide bond and, thus, allows TCR engineering to enhance T-cell functionality. The TCR-targeting vector (Fig 1D) was delivered by AAV6, previously demonstrated to transduce human T cells (Eyquem et al, 2017; Hale et al, 2017; MacLeod et al, 2017). Edited cells expressed the exogenous TCR on their surface after RNP electroporation and subsequent AAV6 transduction, as determined by flow cytometry analysis (Fig 1E). The knock-in efficiency was on average 18% (Fig 1F). This confirms that the correctly integrated transgene can be expressed without an exogenous promoter by using the endogenous one with a promoter trap construct.

To assess whether the introduced transgene was integrated into the TRAC locus as expected, genomic DNA was analyzed with ddPCR. For this, we designed two assays amplifying a region spanning either the whole left (LHA) or the whole right homology arm (RHA), with primer-binding sites in genomic regions and within the transgene. The targeted transgene integration efficiency was 27% (Fig 1G and H).

Previous studies demonstrated that gene disruption of the endogenous TCR locus and subsequent lentiviral introduction of an exogenous TCR result in higher surface expression of the transgenic TCR and decreased off-target reactivity (Provasi et al, 2012). To clarify whether our knock-in approach also exchanged the surface-expressed TCR instead of just adding an additional TCR, thus creating double-positive T cells, we compared the double-positive rate of gene-edited T cells (KI-TRAC-TCR T cells) with that of retrovirally transduced cells (RV-TCR T cells). In KI-TRAC-TCR T cells, the portion of cells expressing both the endogenous human (TCRh) and the exogenous murinized TCR (TCRmu) was significantly decreased (Fig 1E and I). Together, these data demonstrate that RNP electroporation and subsequent AAV6 transduction facilitate the targeted integration of the exogenous TCR in the TRAC locus, thus placing it under endogenous transcriptional control. Furthermore, the reduced double-positive rate in KI-TRAC-TCR cells strongly suggests that the knock-in of a TCR into the TRAC locus not only introduces the exogenous but also replaces the endogenous TCR.

### KI-*TRAC*-TCR T cells specifically recognize and lyse tumor cells in vitro

We set out to assess whether the observed surface expression of the introduced TCR also specifically redirects the edited T cells and whether the editing process has a negative impact on functional capacities of redirected T cells. For this, the edited T cells were co-incubated together with ML-2 acute myeloid leukemia (AML) cells that endogenously present the myeloperoxidase peptide, and which

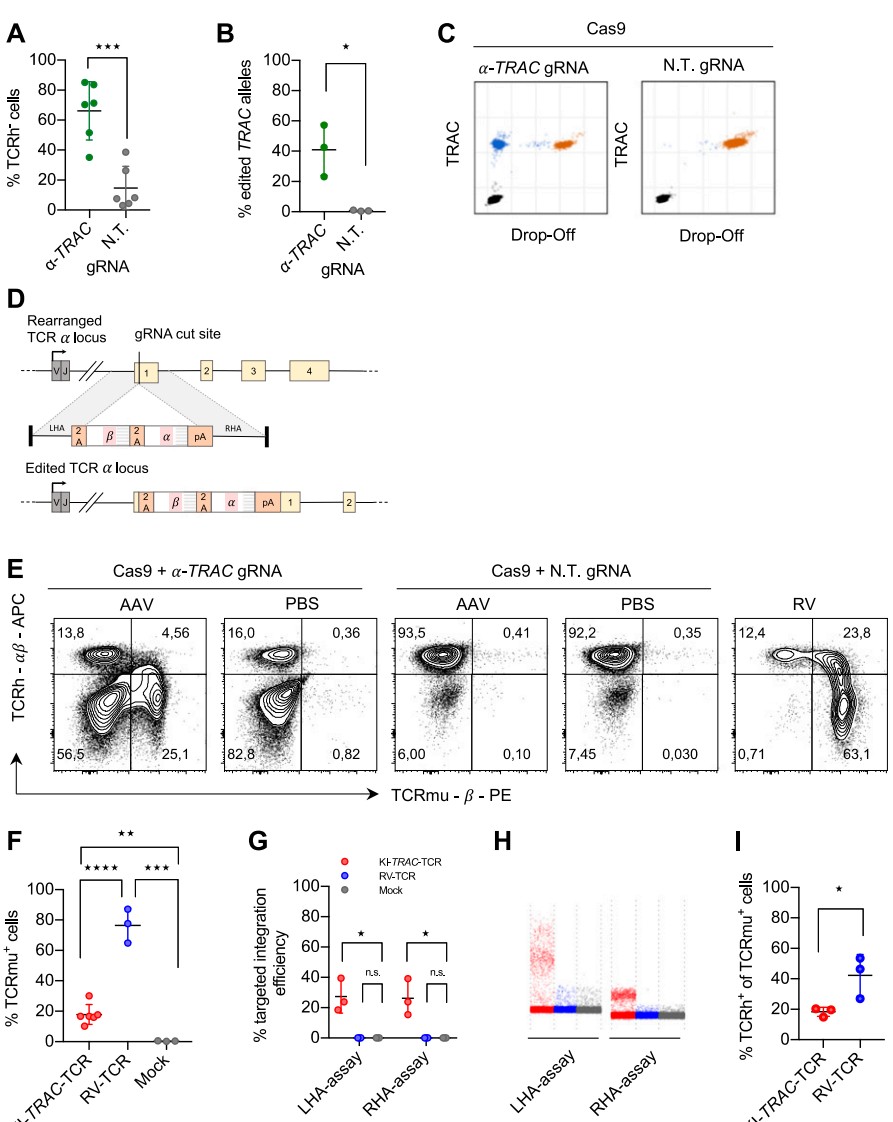

**Figure 1. CRISPR-Cas9- and AAV-mediated TCR replacement.**
**(A)** Flow cytometry analysis of primary human CD8 T cells electroporated with RNPs with an $\alpha$-*TRAC* gRNA or a non-targeting (N.T.) gRNA at day 7 after electroporation (data represent three donors in two independent experiments, $n = 6$). **(B)** ddPCR quantification of the percentage of edited *TRAC* alleles on day 7 ($n = 3$ donors) with 10 ng genomic DNA input. **(C)** Representative ddPCR plots are shown. x and y axes show fluorescence intensity (arbitrary units). **(D)** Schematic representation of the human *TRAC* locus (top), the recombinant AAV6 targeting construct encoding the exogenous TCR (middle) and the successfully edited *TRAC* locus (bottom). LHA, about 900-bp-long left homology arm; RHA, about 900-bp-long right homology arm. **(E)** Representative FACS plots of primary CD8 T cells electroporated with $\alpha$-*TRAC* or N.T. gRNA and transduced with AAV (MOI = $10^6$) or PBS or $\gamma$-retrovirally transduced on day 7 after electroporation or transduction. Axes use biexponential scaling. Graphs are 10% contour plots with outliers displayed. **(F)** Flow cytometry analysis of KI-*TRAC*-TCR cells (data represent three donors in two independent experiments, $n = 6$), $\gamma$-retrovirally ($n = 3$ donors), or mock-transduced cells ($n = 3$ donors). **(G)** ddPCR quantification of the targeted integration efficiency with assays spanning the left (LHA-assay) or right homology arm (RHA-assay). **(H)** Representative ddPCR plots are shown. y axis shows fluorescence intensity (arbitrary units). **(I, F)** Flow cytometry analysis as in (F) ($n = 3$ donors). Asterisks indicate statistical significance as determined by two-tailed unpaired $t$ test. See also Fig S1.

were transgenic for HLA-B7 or control HLA-B15 (Mall et al, 2016). KI-*TRAC*-TCR T cells lysed the tumor cells expressing HLA-B7, whereas not lysing the HLA-B15-bearing control cells (Fig 2A). This demonstrates that the targeted integration of a TCR in the TRAC locus specifically redirects T cells against a defined antigen and that the edited T cells are able to lyse recognized tumor cells in vitro.

Whereas the portion of TCRmu-expressing cells was significantly higher in RV-TCR cells and retrovirally transduced *TRAC*-knockout T cells (RV-TCR TCRendo⁻ T cells) compared with TCR knock-in cells, all modified T cells lysed target cells equally well in vitro at different T cell to tumor ratios, after adjustment of effector to target ratios according to TCRmu⁺ cells (Figs 2B and 1E). The differently genetically modified T cells also showed no marked difference in IFN-γ secretion (Fig 2C). These findings demonstrate that our approach functionally redirects T cells against defined antigens. During the editing procedure, T cells maintain their ability to lyse target cells and secrete IFN-γ.

## KI-*TRAC*-TCR T cells potently reject xenograft tumors in vivo

To explore whether edited T cells also defeat cancer cells in vivo, they were adoptively transferred into NSG mice bearing the subcutaneous AML xenograft ML-2 tumors. Over 3 d, a total of $2 \times 10^7$ TCRmu⁺ KI-*TRAC*-TCR T cells were adoptively transferred, resulting in a significant decrease of tumor size (Fig 3A). When the experiment was terminated on day 7 after the first T-cell dose, most tumors were no longer macroscopically detectable. The tissue at the tumor injection site was harvested, and upon histological examination, only small tumor remnants were visible (Fig 3B). Immunohistochemistry staining for CD3 revealed a massive infiltration of human T cells in the tumor remnants (Fig 3C). When compared with mice treated with RV-TCR T cells or RV-TCR TCRendo⁻ T cells, no notable differences were observed (Figs 3A, D–G and S2). In mice treated only with mock-transduced T cells, on the other hand, we detected large tumors without T-cell infiltration (Fig 3A, H–K). These findings demonstrate

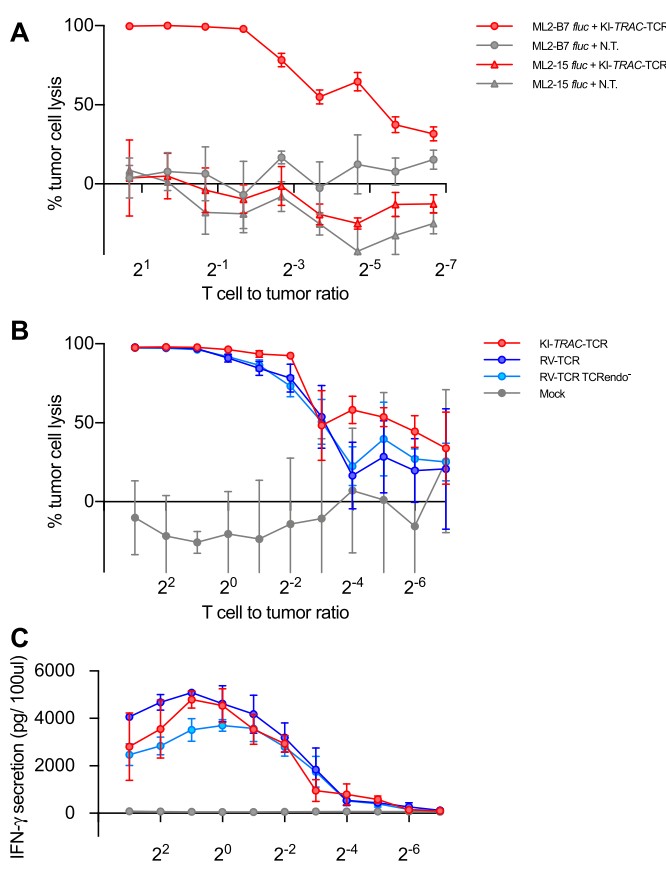

**Figure 2. Targeted integration of a TCR into the *TRAC* locus redirects T-cell specificity.**
**(A)** Cytotoxic lysis of firefly luciferase (*fluc*)–expressing ML2 cells expressing HLA-B7 or HLA-B15 by KI-*TRAC*-TCR cells or unedited T cells at indicated T cell to tumor cell ratios after 24 h of co-incubation assessed by luminescence (*n* = 3 technical replicates of 1 donor). **(B)** *fluc* cytotoxicity assay comparing KI-*TRAC*-TCR cells, RV-TCR, RV-TCR TCRendo⁻, and mock cells after 24 h of co-incubation with ML2-B7 *fluc* at indicated T cell to tumor ratios. T cell to tumor ratios were calculated based on the portion of TCRmu⁺ cells, and equal numbers of TCRmu⁺ cells of each condition were used for co-incubation (*n* = 3 technical replicates of two donors). **(C)** IFN-γ ELISA of the supernatant.

that redirected T cells function very effectively within an in vivo mouse model. The T cells were able to migrate and home to the tumor, lyse the tumor cells within the tumor microenvironment, and reject the tumor in this mouse model of AML within 7 d.

## Discussion

Here, we demonstrate that the specificity of T cells can be redirected by targeting an engineered TCR to the *TRAC* locus using CRISPR-Cas9 RNPs and AAV6 to functionally replace the endogenous TCR with an exogenous, tumor-reactive TCR. The edited cells specifically lyse tumor cells in vitro and control tumors in vivo in a xenograft model of AML.

The observed rate of transgene expression is higher than the previously reported knock-in rate for a TCR into the *TRAC* locus

using dsDNA as a donor template (Roth et al, 2018), even though the introduced TCR construct was 2.3-kb long and contained the complete α- and β chains, therefore permitting TCR engineering approaches to modify the entire TCR α gene. This emphasizes the attractivity of AAV6 as donor templates for HDR-dependent integration. The specific integration of the transgene into the intended locus is consistent with the targeted locus amplification–sequencing profiles presented by previous studies, when targeting transgenes to the *TRAC* locus (Eyquem et al, 2017; Roth et al, 2018). These data confirm that HDR is a reliable mechanism to specifically introduce defined TCR transgenes into the *TRAC* locus. Thus, this approach is suitable to reduce the risk of nonspecific integration in a different locus which could transform edited cells. The use of AAV6 as an HDR donor template is especially attractive as its genome is composed of single-strand DNA. Previous studies have demonstrated that single-strand DNA donor templates integrate more specifically at the intended target site and, therefore may enhance safety of this approach in comparison with double-strand DNA donor templates, which also integrate in an HDR-independent manner at other sites of double-strand breaks (Murnane et al, 1990; Suzuki et al, 2016; Roth et al, 2018). This is important because CRISPR-Cas9 is known to also induce off-target double-strand breaks, into which a transgene could aberrantly integrate (Zhang et al, 2015). Nonetheless, even if there is a small percentage of off-target integration, our HDR-dependent integration approach demonstrates greatly enhanced specificity compared with semi-randomly integrating vectors.

The targeted integration resulted in a significantly reduced proportion of cells expressing the endogenous TCR compared with randomly integrated TCR genes using a retrovirus for genetic transfer. Therefore, the transfer of T cells expressing mixed TCRs can be significantly reduced while providing equal efficacy in vitro as well as in vivo when targeted integration is used instead of retroviral transduction, as shown by our experiments with equilibrated numbers of TCRmu⁺ cells. Thus, this technique provides an important safety advantage compared with other non-targeted gene transfer approaches using viruses or transposons, given that expression of mixed TCR may result in potential harmful toxicities as described in an animal model (Bendle et al, 2010).

Previous studies placing a tumor-reactive receptor under transcriptional control of the *TRAC* locus used xenograft mouse models in which tumor burden was reduced by adoptive T-cell transfer, but the tumors could not be fully removed. In this setting, they were able to demonstrate that edited T cells using the endogenous transcriptional regulation showed better antitumor activity, and in the case of CAR knock-in, a significantly prolonged median survival compared with retrovirally transduced T cells (Eyquem et al, 2017; Roth et al, 2018). In our experiments, we were able to show a strong and fast tumor rejection potential by redirected KI-*TRAC*-TCR T cells, albeit this was not superior in comparison with RV-TCR T cells or RV-TCR TCRendo⁻ T cells. This could be attributed to the fact that we adoptively transferred a higher number of TCR T cells and used a tumor rejection model, where T cells eliminated the tumor within a short period of time. Thus, T-cell exhaustion may not play a major role in our model compared with the models used in previous publications (Eyquem et al, 2017; Roth et al, 2018). Nevertheless, KI-*TRAC*-TCR T cells

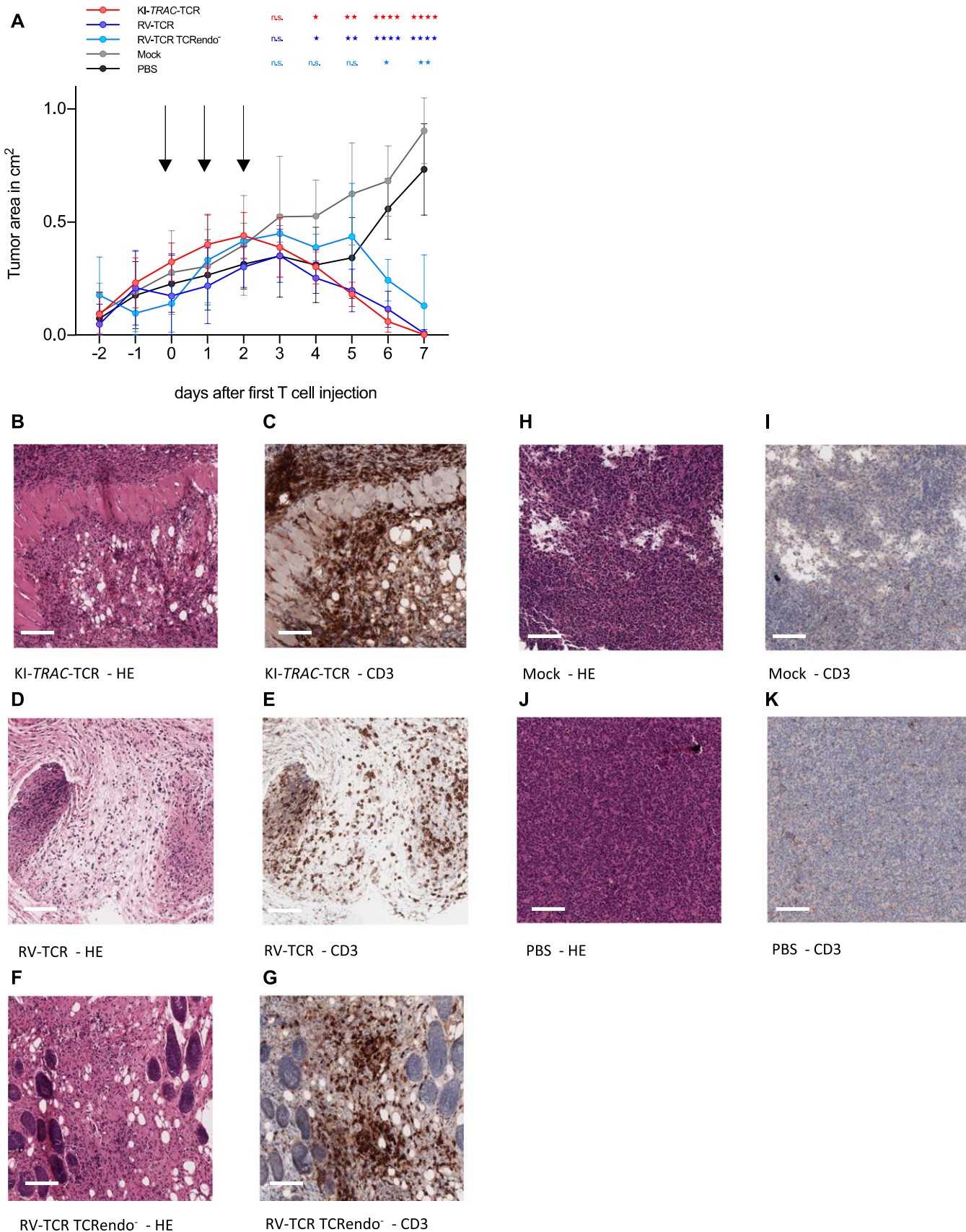

KI-*TRAC*-TCR  - HE

KI-*TRAC*-TCR  - CD3

Mock  - HE

Mock  - CD3

RV-TCR  - HE

RV-TCR  - CD3

PBS  - HE

PBS  - CD3

RV-TCR TCRendo⁻  - HE

RV-TCR TCRendo⁻  - CD3

showed non-inferiority harboring all the advantages of specific integration. Our approach provides a base for in-depth comparisons with T cells modified by targeted integration using dsDNA as well as non-targeted integration by viruses and transposons with respect to subtle differences in safety as well as short- and long-term antitumor immunity.

The method described here can easily be implemented on a laboratory scale and enables researchers to further investigate TCR biology. The role of the regulatory transcriptional network on T-cell functionality and exhaustion in particular can further be addressed. In addition, the targeted integration of TCRs could potentially be adopted for clinical translation as both elements—CRISPR-Cas9 and adeno-associated viruses—are already applied in clinical trials or are even clinically approved. This could lead the field of adoptive T-cell transfer away from randomly integrating vectors and pave the way for new cellular therapies.

# Materials and Methods

### gRNA

The gRNA targeting the first exon of the TRAC locus was designed with the WTSI Genome Editing tool (Hodgkins et al, 2015). The non-targeting gRNA was previously described (Doench et al, 2014). Both RNAs were synthesized by Integrated DNA Technologies as crRNAs and subsequently duplexed with tracrRNAs. α-TRAC gRNA target sequence: 5′-TCTCTCAGCTGGTACACGGC-3′; non-targeting gRNA: 5′-GTATTACTGATATTGGTGGG-3′.

### RNP production

CRISRP-Cas9 RNPs were assembled from the crRNA:tracrRNA duplexes and Alt-R Sp Cas9 Nuclease (Integrated DNA Technologies) according to the manufacturer's recommendations. Briefly, crRNA and tracrRNA were equimolarly mixed and resuspended in IDTE buffer at a concentration of 44 $\mu$M. This mix was heated to 95°C for 5 min and let cool down to room temperature. crRNA:tracrRNA duplexes were mixed with 36 $\mu$M Cas9 enzyme and incubated at room temperature for 20 min. RNPs were prepared on the day of electroporation and stored at 4°C.

### Isolation of primary human CD8 T cells for gene targeting

Blood from healthy donors was acquired with informed consent according to the Helsinki Declaration and the local ethical board. PBMCs were isolated by density-gradient centrifugation with Ficoll/Hypaque (Biochrom) and frozen in FCS supplemented with 10% DMSO. For genetic modification, the cells were thawed and CD8

T cells were isolated via negative selection with the Dynabeads Untouched Human CD8 T Cells kit (Thermo Fisher Scientific).

### Primary human T-cell culture and ML2 cell line

T cells were kept at a density of ~$10^6$ cells per ml cell culture medium. The T-cell medium consisted of RPMI 1640 (Invitrogen), 1× penincillin/streptomycin (Invitrogen), 5% FCS, 5% human serum, 1 mM sodium pyruvate (Invitrogen), 2 mM L-glutamine (Invitrogen), 10 mM nonessential amino acids (Invitrogen), 10 mM Hepes (Invitrogen), and 16 $\mu$g/ml gentamycin (Biochrom). Human IL-7 and human IL-15 (both PeproTech) were added to the medium to a final concentration of 5 ng/ml each and replenished when fresh culture medium was added to the cells every 2–3 d. The AML cell line ML2 (The CABRI consortium) was retrovirally transduced with genes encoding firefly luciferase (fluc) or HLA-B7. Mycoplasma contamination status was regularly tested.

### Target vector construction

The target vector pAAV-L900-PT-TCR2.5D6omc-R900 was cloned based on the plasmid pAAV-CMV-eGFP. The construct bears two inverted terminal repeats (ITRs) based on AAV2, which are cis-acting sequences needed for successful viral packaging of the sequences between the ITRs. Primers were designed to amplify homology arms covering 900-bp upstream or downstream of the intended double-strand break that has been induced by CRISPR-Cas9. The primers were designed in such a way that the protospacer adjacent motif was removed to prevent additional recognition by Cas9, and restriction enzyme–binding sites were added for subsequent restriction–ligation cloning. The homology arms were amplified from genomic DNA and cloned in pCR-Blunt plasmids (Thermo Fisher Scientific) and afterwards cloned between the ITRs of pAAV-CMV-GFP. The TCR construct used for the targeted integration in the TRAC locus was designed based on the previously described sequence of TCR2.5D6 (Klar et al, 2014). Briefly, in this sequence, the constant regions of the TCR α- and β chain are replaced by murine sequences in which an additional cysteine bond is introduced. The α- and β chains are connected by a P2A element to permit bicistronic expression from one promoter. To place the integrated TCR under the transcriptional control of the endogenous promoter of the TCR α chain, a P2A element was placed upstream of the TCR construct (Eyquem et al, 2017). This promoter trap was designed in such a way that it is located in frame within the first exon of the TRAC locus after successful integration. The construct was synthesized as GeneArt Strings DNA Fragment (Invitrogen). Subsequently, the integration cassette was cloned between the homology arms in the pAAV plasmid. The following primers were used for the amplification of the homology

---

**Figure 3. T cells potently reject tumors in vivo independent of the way of genetic modification applied.**
**(A)** Tumor area of subcutaneous ML2-B7 fluc tumors in NOD.CG-Prkdc$^{scid}$ IL2rg$^{tm1Wjl}$/SzJ (NSG) mice after adoptive transfer of 2 × $10^7$ TCRmu$^+$ KI-TRAC-TCR cells (n = 8), RV-TCR cells (n = 7), RV-TCR TCRendo$^-$ (n = 3), mock T cells (n = 6) derived from one donor, or PBS (n = 6). Arrows indicate injection of 6.67 × $10^6$ TCRmu$^+$ T cells in 200 $\mu$l PBS. Asterisks indicate statistical significance as determined by multiple t tests corrected for multiple comparisons using the Holm–Sidak method. **(B–K)** Representative HE and CD3 stains of tissue derived from mice treated with KI-TRAC-TCR cells (B, C), RV-TCR cells (D, E), RV-TCR TCRendo$^-$ (F, G), mock T cells (H, I), or PBS (J, K). Scale bars, 100 $\mu$m. See also Fig S2.

arms: LHA_for: 5'-GGCGCGCCCACTAAGGAAAAG-3'; LHA_rev: 5'-ACTAGTGTCAGGGTTCTGGATATCTG-3'; RHA_for: 5'-GGTAACCGTA-TACCAGCTTCGAGACTCTAAATCCAGTGACA-3'; RHA_rev: 5'-CGGTCC-GCAAGTAGCATTTCTTCAGAG-3'.

## Recombinant AAV6 production

Recombinant adeno-associated viruses of the serotype 6 were produced with the triple transfection method as described previously (Ziegler et al., 2017). Briefly, the packaging cell line U293 was transfected with the vector pAAV-L900-PT-TCR2.5D6omc-R900, a plasmid encoding the cap sequences of AAV6 and rep AAV2 sequences and the helper plasmid δ F6 (Puresyn) using PEI Max (Polysciences). After 72 h, the cells were harvested and the virus was purified by iodixanol-gradient centrifugation.

The virus was further purified by a gravity flow size exclusion purification using Sepharose G100 SF resin (Sigma-Aldrich) in Econopac columns (Bio-Rad Laboratories) using the protocol kindly provided by the Salk Institute for Biological Studies.

The virus was concentrated in DPBS using Amicon Ultra-15 Centrifugal Filter Units (Merck) and stored at 4°C. Viral titer was quantified by quantitative PCR as described previously (Aurnhammer et al, 2012).

## Gene targeting

Primary human T cells were activated with Dynabeads Human T-Activator CD3/CD28 (Thermo Fisher Scientific) according to the manufacturer's recommendations at a bead-to-cell ratio of 1:1 in the presence of 30 U/ml human IL-2 (PeproTech) for 72 h.

After bead removal, $10^6$ cells were washed in DPBS and resuspended in 11 $\mu$l resuspension buffer R of the Neon electroporation kit (Invitrogen). 1 $\mu$l of the RNP mix was added to the cells immediately before electroporation. 10 $\mu$l of the mixture was electroporated with the Neon electroporation device at 1,600 V, 10 s with three pulses. The cells were transferred to 1 ml of prewarmed T-cell medium. Immediately afterwards, the virus solution was added at a multiplicity of infection (MOI) of $10^6$ viral genomes per cell. For controls (TCRendo$^-$ T cells), the same volume of DPBS was added.

## γ-Retroviral transduction of T cells

Primary human T cells were transduced with the pMP71-TCR2.5D6 construct previously described (Klar et al, 2014) according to a protocol previously published (Liang et al, 2010). The embryonal kidney cell line 293Vec-RD114 (BioVec Pharma) was used as the packaging cell line.

## ddPCR

Genomic DNA was isolated from the cells with the DNeasy Blood & Tissue kit (QIAGEN), following the manufacturer's instructions. DNA was quantified using a Qubit 2.0 fluorometer (Thermo Fisher Scientific). ddPCR analysis was performed on a QX200 ddPCR system with automatic droplet generation (Bio-Rad Laboratories). Reactions were carried out in ddPCR 96-well plates (#12001925; Bio-Rad Laboratories) and were performed in duplicate in a reaction volume of 21 $\mu$l using the ddPCR Supermix for probes (no UTP,

Bio-Rad Laboratories) and with a template input of 10 ng (LHA/RHA and Drop-Off/TRAC) or 100 ng DNA (EIF2C1 and Drop-Off/TRAC) per reaction. The locked nucleic acid probe and the EIF2C1 primers and probe were synthesized by Integrated DNA Technologies, and all other primers and hydrolysis probes were synthesized by Sigma-Aldrich: DO-TRAC-for: 5'-CTTGTCCATCACTGGCATCT-3'; DO-TRAC-rev: 5'-CGGTGAATAGGCAGACAGAC-3'; TRAC-probe: 5'-[6FAM]AGCCTGGG-TTGGGGCAAAGAGGG[BHQ1] -3', LNA-Drop-Off: 5'-/5HEX/CCCTGC+C+G+T+GTA/3IABkFQ/-3', LHA-for: 5'-CCCCAACATGCTAATCCTCC-3'; LHA-rev: 5'-ACAGCAGTCCCAGAGACATA-3'; LHA-probe: 5'-[6FAM]CC-CCACAGAGCCCCGCCCT[BHQ1]-3', RHA-for: 5'-TCAGGTGATCTACCCAC-CTT-3'; RHA-rev: 5'-GTCTACCCTCTCATGGCCTA-3'; RHA-probe: 5'-[HEX]CAGGGCCGGGTCACAGGGCC[BHQ1]-3', and EIF2C1-for: 5'-CCTG-CCATGTGGAAGATGAT-3'; EIF2C1-rev: 5'-GAGTGTGGTCACTGGACTTG-3'; EIF2C1-probe: 5'-/5HEX/ACCAGTCTG/ZEN/TGCGCCCTGCCA/3IABkFQ/-3'. Primers were used at a final concentration of 900 nM and probes at 250 nM. Genomic DNA derived from leukocytes of healthy subjects was used as negative control, Sanger-sequenced plasmid DNA as positive control, and purified, nuclease-free water as no-template control in all ddPCR runs. The thermal cycling protocol was as follows: 10 min at 95°C, 40 cycles of 30 s at 94°C, 1 min at 60°C for annealing/extension, and one final step of 10 min at 98°C. For the LHA/RHA PCR, resulting in an amplicon of 2,337 bp, annealing/extension time was increased to 3 min. In-PCR digestion was performed with HindIII (LHA/RHA), HaeIII (EIF2C1), or MseI (Drop-Off/TRAC) in all cases. The plates were read on a QX200 droplet reader (Bio-Rad Laboratories) to determine droplet fluorescence intensity.

## ddPCR data analysis

Droplets were manually assigned to double-negative, single-positive, or double-positive droplet clusters after visual inspection in QuantaSoft v1.7.4 (Bio-Rad Laboratories). Raw droplet fluorescence intensity values including cluster assignments were exported from QuantaSoft. Custom scripts were used to import the intensity values into R (version 3.4.4; http://www.r-project.org), to generate plots, and to quantify sample concentrations. Target concentrations were calculated for each well from the number of positive droplets Np and negative droplets Nn and the average droplet volume V = 0.85 nl based on Poisson distribution statistics using the formula c = (ln(Np + Nn) − ln(Nn))/V, where ln is the natural logarithm. Following Mock et al (2016), rain droplets were considered negative because they represent mutated alleles. Gene-editing frequencies were calculated as the ratio of edited over edited plus wild-type allele concentrations.

## In vitro functional assays

T cells were incubated together with $10^4$ firefly luciferase expressing ML-2 HLA-B7 *fluc* or ML-2 HLA-B15 *fluc* cells at indicated ratios in 200 $\mu$l cytokine-free T-cell medium. T cell to tumor ratios were calculated based on the portion of TCRmu$^+$ cells and equal numbers of TCRmu$^+$ cells of each condition were used for co-incubation. After 24 h, IFN-γ secretion into the supernatant was analyzed by ELISA (BD) according to the manufacturer's instructions. Cytotoxicity was assessed by adding 100 $\mu$l luciferase

substrate solution consisting of DPBS (Thermo Fisher Scientific), 10% FCS, and 150 μg/ml D-luciferin (Gold Biotechnology) to the cell pellet. Immediately afterwards, luminescence was determined with a VICTOR Multilabel Plate Reader (PerkinElmer). Tumor cell lysis was determined as tumor lysis = (1 − (luminescence of sample/luminescence of untreated tumor cells) × 100). Negative lysis indicates cell growth compared with the untreated tumor cell controls.

### In vivo mouse model

NOD.CG-*Prkdc*^*scid* *IL2rg*^*tm1Wjl*/SzJ (NSG; The Jackson Laboratory) were maintained according to the institutional guidelines and approval of the local authorities. $10^7$ ML2-B7 *fluc* cells were subcutaneously injected in the right flank of 8- to 12-wk-old mice. On day 8 after tumor cell injection, the mice were randomly assigned to the different treatment groups. On days 8, 9, and 10, edited T cells, retrovirally transduced T cells, retrovirally transduced TCR knockout T cell, mock-transduced T cells, or PBS were intravenously injected. A total of $2 \times 10^7$ TCRmu$^+$ T cells were injected in three doses of 200 μl PBS each. As the transduction efficiency of the retrovirally treated cells was higher than that of the edited cells, mock-transduced cells were used to adjust the percentage of TCRmu$^+$ T cells in the treatment groups (≈20%). Tumors were measured daily with a caliper, and tumor area was calculated as A = length × width.

### Flow cytometry

The cells were stained with combinations of the following antibodies: anti-human TCR$\alpha$/$\beta$-PE (BW242/412; Miltenyi Biotec), anti-mouse TCR $\beta$ chain-APC (H57-597; BD), and anti-hCD3-PE (UCHT1; BD). In brief, 100,000 cells were washed in FACS buffer (PBS + 10% FCS) and blocked with 50 μl of human serum for 10 min on ice. The cells were washed again with FACS buffer, stained with the respective antibodies, and incubated for 20 min on ice. Subsequently, the cells were washed with FACS buffer and fixed in 1% PFA. The cells were analyzed with an LSRII Flow Cytometer (BD) or a CytoFLEX S (Beckman Coulter), and data were analyzed with FlowJo 10 (FlowJo LLC).

### Histology and immunohistochemistry

Directly after euthanasia of the mice, tissues were fixed in 4% neutral-buffered formalin solution for at least 48 h, dehydrated under standard conditions (ASP300S; Leica Biosystems) and embedded in paraffin. Serial 2-μm-thin sections prepared with a rotary microtome (HM355S; Thermo Fisher Scientific) were collected and subjected to histological and immunohistochemical analysis. Hematoxylin-eosin staining was performed on deparaffinized sections.

Immunohistochemistry was performed with a BenchMark XT automated stainer (Ventana) with an antibody against CD3 (103R-95, MRQ-39), using the ultraVIEW DAB Detection kit (all reagents were from Ventana). Briefly, the tissue sections were deparaffinized with EZ Prep at 75°C and 76°C, heat-pretreated in Cell Conditioning 1 (CC1) solution for antigen retrieval at 76°C–100°C and then incubated with the primary antibody diluted in antibody diluent 1:500

for 32 min at 37°C after inactivation of the endogenous peroxidase using a UV inhibitor for 4 min at 37°C. The slides were incubated with an HRP Universal Multimer for 8 min. Antibody binding was detected using DAB as chromogen and counterstained with hematoxylin for 10 min with subsequent bluing in bluing reagent for 10 min. The slides were then manually dehydrated by alcohol washes of increasing concentrations (70%, 96%, and 100%) and xylene and cover-slipped using Pertex mounting medium (00801; Histolab). The stained slides were evaluated by an experienced certified pathologist (K Steiger) using a BX53 stereomicroscope (Olympus) and scanned using a slide scanner (AT-2; Leica Biosystems). Representative images were collected using Aperio Imagescope software (version 12.3; Leica Biosystems).

### Statistics

All statistical analyses were performed as indicated in the figure legends using GraphPad Prism 7. Error bars in all figures represent mean ± SD. *$P$ = 0.01–0.05, **$P$ = 0.001–0.01, ***$P$ = 0.0001–0.001, and ****$P$ < 0.0001.

## Supplementary Information

## Acknowledgements

We thank Roland Rad (TUM), Rupert Öllinger (TUM), Marc Schmidt-Supprian (TUM), Dieter Saur (TUM), and Roland Schmid (TUM) for their helpful discussion and comments. We thank all lab members for their valuable support. This research was supported by the Deutsche Forschungsgemeinschaft (SFB824/C10 to AM Krackhardt; SFB824/Z02 and SFB1335/Z01 to K Steiger and SFB 1054/B01, SFB 1335/P01 and P08, TRR 237/A10 to J Ruland), the European Research Council (FP7, grant agreement no. 322865) to J Ruland, and the TUM Medical Graduate Center to JJ Albers.

### Author Contributions

JJ Albers: conceptualization, data curation, formal analysis, investigation, visualization, methodology, and writing—original draft.
T Ammon: supervision and writing—review and editing.
D Gosmann: investigation and writing—review and editing.
S Audehm: investigation and writing—review and editing.
S Thoene: investigation and writing—review and editing.
C Winter: investigation and writing—review and editing.
R Secci: investigation and writing—review and editing.
A Wolf: investigation and writing—review and editing.
A Stelzl: investigation and writing—review and editing.
K Steiger: investigation and writing—review and editing.
J Ruland: investigation and writing—review and editing.
F Bassermann: investigation and writing—review and editing.
C Kupatt: investigation and writing—review and editing.
M Anton: investigation and writing—review and editing.
AM Krackhardt: conceptualization, supervision, and writing—review and editing.

## Conflict of Interest Statement

The authors declare that they have no conflict of interest.

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
