## [Reviewer comments · Life Science Alliance]

Life Science Alliance

Gene editing enables T cell engineering to redirect antigen specificity for potent tumor rejection

Julian Albers, Tim Ammon, Dario Gosmann, Stefan Audehm, Silvia Thoene, Christof Winter, Ramona Secci, Anja Wolf, Anja Stelzl, Katja Steiger, Juergen Ruland, Florian Bassermann, Christian Kupatt, Martina Anton, and Angela Krackhardt

DOI: <https://doi.org/10.26508/lsa.201900367>

Corresponding author(s): Angela Krackhardt, Technische Universität München, Klinikum rechts der Isar

Review Timeline:	Submission Date:	2019-03-05
	Editorial Decision:	2019-03-06
	Revision Received:	2019-03-06
	Accepted:	2019-03-07

Scientific Editor: Andrea Leibfried

Transaction Report:

Please note that the manuscript was previously reviewed at another journal and the reports were taken into account in the decision-making process at Life Science Alliance. Since the original reviews are not subject to Life Science Alliance's transparent review process policy, the reports and author response cannot be published.

March 6, 2019

RE: Life Science Alliance Manuscript #LSA-2019-00367-T

Dr. Angela M Krackhardt
Technische Universität München, Klinikum rechts der Isar
IIIrd Medical department
Ismaningerstr. 22
Munich 81675
Germany

Dear Dr. Krackhardt,

Thank you for transferring your manuscript entitled "Gene editing enables T cell engineering to redirect antigen specificity for effective tumor rejection" to Life Science Alliance. Your manuscript was previously reviewed at another journal, and the editors transferred those reports to us with your permission.

The reviewers thought that your work is robust and that it corroborates and extends previous findings. However, they also thought that the broader conceptual novelty provided remains somewhat low. This is not a concern for publication in Life Science Alliance, and we would thus like to invite you to submit a revised version to us for publication here. The individual reviewer points should get addressed by altering the discussion in your manuscript.

A. FINAL FILES:

B. MANUSCRIPT ORGANIZATION AND FORMATTING:

Thank you for your attention to these final processing requirements.

Sincerely,

March 7, 2019

RE: Life Science Alliance Manuscript #LSA-2019-00367-TR

Dr. Angela M Krackhardt
Technische Universität München, Klinikum rechts der Isar
IIRD Medical department
Ismaningerstr. 22
Munich 81675
Germany

Dear Dr. Krackhardt,

Thank you for submitting your Resource entitled "Gene editing enables T cell engineering to redirect antigen specificity for potent tumor rejection". I appreciate the introduced changes that address the concerns of the previous reviewers and it is a pleasure to let you know that your manuscript is now accepted for publication in Life Science Alliance. Congratulations on this interesting work.

*****IMPORTANT:** If you will be unreachable at any time, please provide us with the email address of an alternate author. Failure to respond to routine queries may lead to unavoidable delays in publication.*******

DISTRIBUTION OF MATERIALS:

Again, congratulations on a very nice paper. I hope you found the review process to be constructive and are pleased with how the manuscript was handled editorially. We look forward to future exciting

submissions from your lab.

Sincerely,
